



# The 2022-2023 snow drought in the Italian Alps doubled glacier contribution to summer streamflow

Martina Leone[1,2,3], Francesco Avanzi[2], Umberto Morra di Cella[2,4], Simone Gabellani[2], Edoardo Cremonese[2], Michel Isabellon[2], Paolo Pogliotti[4], Riccardo Scotti[5], Andrea Monti[5], Luca Ferraris[2,3], and Roberto Colombo[1]

[1]Department of Earth and Environmental Sciences, University of Milano-Bicocca, LTDA, Piazza della Scienza 1, 20126 Milan, Italy
[2]CIMA Research Foundation, Via Armando Magliotto 2, Savona 17100, Italy
[3]Dipartimento di informatica, bioingegneria, robotica e ingegneria dei sistemi - DIBRIS, Università di Genova, Genova, Italy
[4]Climate Change Unit, Environmental Protection Agency of Aosta Valley, Loc. La Maladière, 48-11020 Saint-Christophe, Italy
[5]Servizio Glaciologico Lombardo - Glaciological Service of Lombardy, Italy
**Correspondence:** Martina Leone (martina.leone@cimafoundation.org)

**Abstract.**

We quantified the role of glaciers in mitigating snow-drought impacts on downstream streamflow during the severe 2022-2023 event in the Italian Alps. In order to do so, we compared glacier-melt contribution to streamflow during these years with the 2011-2023 historical period in two catchments, Dora Baltea (Aosta Valley) and Adda (Lombardy). We employed spatially

distributed estimates of glacier melt, snow water equivalent (SWE), air temperature and total precipitation over glaciers from an operational cryospheric model (S3M Italy), and compared these estimates with downstream observations of streamflow at the closure sections of both catchments. Results showed a severe snow water equivalent deficit over glaciers across both catchments and both years (between ∼ -45 % at 4000 m in 2022 and ∼ -75 % at 2000 m a.s.l. during both years), which was largely driven by anomalous air temperatures and seasonal-precipitation patterns (up to +2-3 °C and -73%, respectively). Air-

temperature anomalies displayed a clear signature of orographic-dependent warming, with anomalies at 4000 m a.s.l. that were 1 to 1,5 °C higher than at 2000 m a.s.l.. Glacier contribution to streamflow doubled to tripled during these snow droughts in both catchments, a process that manifested itself through four mechanisms: an earlier-than-usual onset of the glacier melt season, an intensification of glacier melt contribution to streamflow, an earlier-than-usual seasonal peak in glacier melt contribution, and an extension of the glacier melt season. Still, glacier melt contribution to streamflow remained highly sensitive to short-term

meteorological events, such as a sudden drop of temperatures, as well as early/late snowfalls. These results highlight the critical role of glacier melt in maintaining streamflow during severe droughts and emphasize the need to integrate glacier dynamics into water management strategies for alpine areas facing increasingly frequent and intense drought events.



# 1 Introduction

Glaciers play a crucial role in regulating water availability in and from mountainous regions (Viviroli et al., 2007; Immerzeel
et al., 2010). They do so by significantly contributing to streamflow, especially during summer, and thus providing a vital
source of freshwater for human use and ecosystems (Viviroli et al., 2020; Soruco et al., 2015; Barnett et al., 2005; Immerzeel
et al., 2010; Hanus et al., 2024). However, glacier coverage is shrinking at an unmatched rate as the climate warms (Hugonnet
et al., 2021; Zemp et al., 2015; Sommer et al., 2020), driven by decreased snowfall and snow cover (Colombo et al., 2022;
Bozzoli et al., 2024; Marty et al., 2017; Mote et al., 2018; Ngoma et al., 2021) and negative glacier mass balances (Zemp
et al., 2023; Zekollari et al., 2019; Huss et al., 2017). Projections for the European Alps indicate a potential loss of 50% of the
2017 glacier volume by 2050, and up to the complete disappearance by the end of the century, depending on the considered
emission scenario (Zekollari et al., 2019). The supportive and often decisive role of glaciers as global "water towers" is thus
under pressure (Viviroli et al., 2020; Gobiet et al., 2014; van der Wiel et al., 2021; Stahl et al., 2022).

In this context of climate warming, a growing concern in the hydrology of world-wide mountains is the emergence of snow
droughts, that is, periods characterized by a significant lack of snowfall, or a lack of snow accumulation during winters with
near normal winter precipitation but higher-than-usual temperatures (Harpold et al., 2017; Huning and Aghakouchak, 2020;
Hatchett et al., 2022). Low snow accumulation has significant negative consequences: it poses challenges for water management
(Harpold et al., 2017), threatens food security, and harms wildlife and ecosystems (Barsugli et al., 2020). Furthermore, a
deficit of meltwater runoff due to a snow drought can diminish hydropower potential, as observed in Northern Italy during
2022 (Koehler et al., 2022). Finally, by combining dry and warm conditions, snow droughts have the additional potential to
exacerbate glacier shrinkage through prolonged melt seasons (Vargo et al., 2020).

The exact chain of processes linking snow droughts to glacier melt enhancement and downstream water supply deficit during
summer has rarely been characterized and thus remains largely unknown (Van Tiel et al., 2021). Understanding this chain of
processes is urgent, as some early studies have shown that glacier melt can compensate for low flows during droughts in alpine
regions (Van Tiel et al., 2021, 2023). Following these early studies, it appears that glaciers may offset precipitation-driven water
shortages by exceeding normal ice melt (Van Tiel et al., 2021), due to increased meltwater during periods of lack of snow and
heatwaves. This enhanced glacier melt, coupled with lower-than-usual streamflow due to the lack of precipitation, increases
the contribution of glaciers to streamflow during droughts. However, these early studies also indicate that the compensatory
role of glaciers is rarely straightforward, and significantly varies with local factors and hydrological regimes.

Also, early studies on the contributing role of glaciers during streamflow droughts have mostly been in temperate regions,
while elucidating the chain of events at play during these events is particularly important in Mediterranean regions, where
water supply is asynchronous between wet-cold winters and dry-warm summers (Bales et al., 2018). In such climates, glaciers
may seasonally represent a predominant source of water for all sectors and uses, with little to none contribution from summer
precipitation (Ayala et al., 2020). This is the case of the southern side of the European Alps, a region acting as a crucial hinge
between the climate regimes of northern Europe and the Mediterranean. This transitional climate zone is projected to face
increasingly intense and severe droughts under future climate scenarios ((Bednar-Friedl et al., 2022)). Consequently, glaciers





in the southern Alps are becoming increasingly vital for both upstream communities relying on meltwater and downstream economies dependent on consistent water resources (Beniston et al., 2018).

The snow droughts of 2022 and 2023 in the Italian Alps (Colombo et al., 2023) present an opportunity to investigate the
contribution of glaciers to streamflow during snow droughts. 2022 was globally the fifth warmest year on record (World Meteorological Organization, 2023), with Europe experiencing its hottest summer since 1950, and was marked by a persistent high pressure, heatwaves up to +2,5°C above normal (Tripathy and Mishra, 2023), and significant precipitation deficits (Copernicus Climate Change Service, a). Northern Italy was particularly affected by combined drought and heat, leading to a record-low snow water equivalent (SWE) in March 2022, down -70% compared to the reference period (Avanzi et al., 2024). This ex-
treme warmth and lack of snow resulted in unprecedented glacier losses (Cremona et al., 2023; Voordendag et al., 2023) and the most severe streamflow drought in the Po river basin in two centuries (Montanari et al., 2023). 2023 followed as the warmest year globally since 1880 (World Meteorological Organization, 2024), and the second warmest in Europe (Copernicus Climate Change Service, b). While Europe saw varied conditions, northern Italy and the Alps consistently experienced a winter snow drought, followed by a warm and dry summer with prolonged heatwaves (Copernicus Climate Change Service,
b). The standardized snow water equivalent index (SSWEI) for the 2022-2023 season was -2,8 (for a definition of the snow water equivalent index, see Colombo et al., 2023), further underscoring the severity of these two years. Both characterized by intense heat and exceptionally low snowfall accumulation (Avanzi et al., 2022b), 2022 and 2023 demonstrate the growing vulnerability of alpine regions to water scarcity, and are a suitable example for studying the compensatory role of glaciers during snow droughts (Cremona et al., 2023).
Here, we aim to quantify the temporal and spatial patterns of glacier-melt contribution to streamflow in the Italian Alps during the intense 2022 and 2023 snow drought. We used spatially distributed estimates of snow water equivalent and glacier melt from two of Italy's most glacierized regions, Aosta Valley and Lombardy, along with streamflow data, over a period of 13 years (2010-2023), including the most recent 2022-23 snow droughts. We aimed to answer the following two research questions: (i) what are the key mechanisms driving the response of glacier melt to a snow drought? (ii) How much does glacier
melt contribute to river flow during snow droughts compared to average years?

## 2 Study area

The Italian Alps represent a major glacial landscape in southern Europe. Here, we focus on two highly glaciated regions in particular: Aosta Valley and Lombardy (Smiraglia et al., 2015), illustrated in Figure 1.

In Aosta Valley, 4 % of the territory (134 km$^2$) is glacier-covered, encompassing 184 glaciers (Sottozero). For the validation
of glacier-melt simulations (see Section 3), we used available data on the Timorion, Rutor, and Petit Grapillon glaciers, which represent a range of glacial environments within Aosta Valley with available data (see Section 3). The Dora Baltea river flows through this valley, with Tavagnasco being the representative closure section for the basin (Figure 1).

In Lombardy, glaciers cover 73 km$^2$ across 203 glaciers (Bonardi, 2012), that is, about 0,3 % of the total area. Again for validation purposes (see Section 3), we focused on two distinct glacial systems with available data: Ortles-Cevedale, with





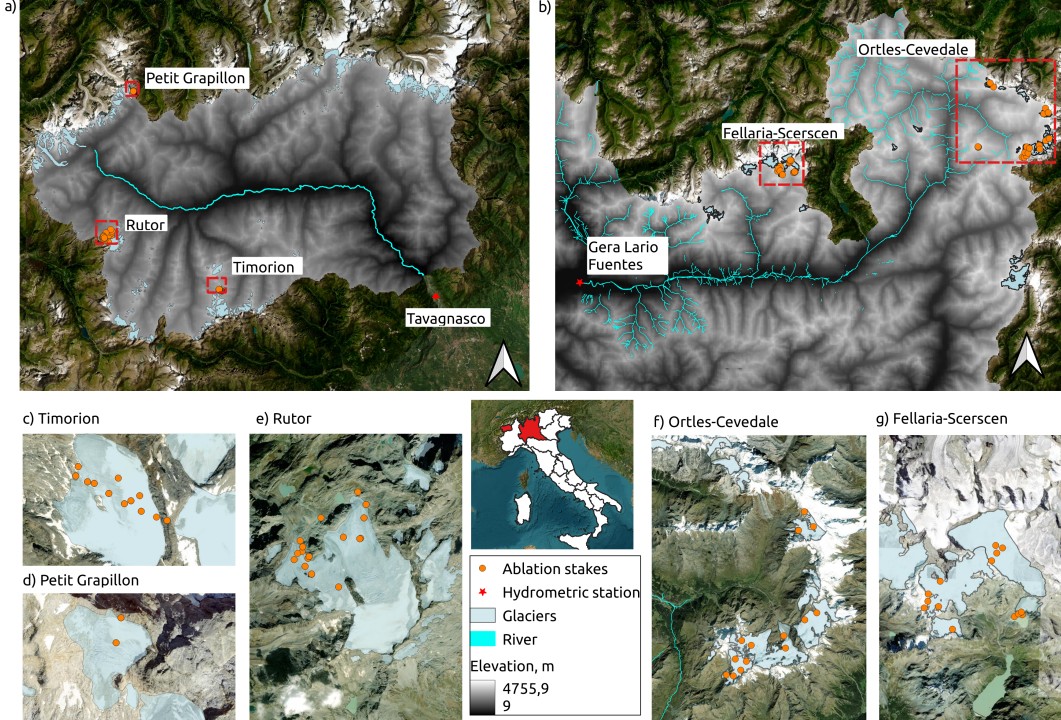

**Figure 1.** Study area. The top left figure represents Aosta Valley (a), while the top right figure represents northern Lombardy (b). The blue lines represent the Dora Baltea and Adda rivers, while the pale blue areas represent glaciers. The orange dots correspond to ablation-stake data used in this study. The red squares refer to various zooms (panels c, d, e, f, g). The red stars represent streamflow-data stations used in this study: Tavagnasco and Gera Lario Fuentes. The digital elevation model of the Italian Alps is shown in grey scale for reference. On the left portion, it is possible to observe a zoom on the following glacier areas: Timorion (c), Petit Grapillon (d) and Rutor (e), all in Aosta Valley; on the right portion, it is possible to observe a zoom on the following glacier areas: Ortles-Cevedale (f) and Fellaria-Scerscen (g), both located in Lombardy. Base map: Esri, Maxar, Earthstar Geographics, and the GIS User Community. Glacier outlines from updated datasets by the Aosta Valley Regional Authority (2019) and the Lombardy Glaciological Service (2021). Digital elevation data from S3M Italy. Map generated using open-source GIS tools.

multiple glaciers such as Alpe Sud, Cedec, Cevedale, Forni, Dosegù and Vitelli, and Bernina, including East and West Fellaria and Scerscen glaciers. These glaciers feed the Adda River, with Gera Lario Fuentes representing the closure section (Figure 1, glaciers represent the 1,77 % of the basin area).



## 3    Data and methods

### 3.1    Data: S3M Italy, glacier ablation stakes, and streamflow

To estimate glacier melt, we used output from the S3M Italy operational chain (Avanzi et al., 2023), which relies on the S3M snow-glacier model (Avanzi et al., 2022a). The model uses an hybrid temperature index and radiation-driven melt approach and is fed by hourly inputs of incoming shortwave radiation, air temperature, total precipitation, and relative humidity. Note that in S3M equations are solved for each pixel with no exchange of mass or energy across pixels, including no wind redistribution.

In the S3M Italy operational chain, S3M works with a 200 m spatial resolution and simulates hourly snapshots of snow water

equivalent (SWE), snow depth, bulk snow density, and glacier melt across the whole of the Italian territory (period: September 2010 to present). Input data are routinely obtained from the database of the Italian Regional Administrations, Autonomous Provinces, and the Italian Civil Protection. Input maps are cropped over the 20 computational domains, each corresponding to one Italian administrative region, originally derived from a 20 m digital elevation model provided by the Italian Institute for Environmental Protection and Research (ISPRA), which was resampled at 200 m resolution using an averaging method.

Besides elevation, S3M Italy employs static glacier maps from the Randolph Glacier Inventory v 6.0. The outputs are maps of snow accumulation and melt, as well as glacier melt on snow-free glacier surfaces. S3M also does not include glacier movement or debris coverage: while the first aspect is likely a second-order approximation considering the fairly short period considered in this study, we will further discuss the implication of neglecting debris coverage in Section 5.

After a calibration in north-western Italy based on minimizing errors with respect to a variety of snow depth data (Avanzi

et al., 2022a), S3M Italy has been further validated with regard to its snow component at the national scale (Avanzi et al., 2023), showing little to no mean bias compared to Sentinel-1-based maps of snow depth, and root mean square errors are of the typical order of 30–60 cm and 90–300 mm for in situ, measured snow depth and snow water equivalent, respectively. Estimates of peak snow water equivalent by S3M Italy are also well correlated with annual streamflow at the closure section of 102 basins across Italy (0.87), with ratios between peak water volume in snow and annual streamflow that are in line with expectations for

this mixed rain–snow region (22 % on average and 12 % median). On the other hand, no specific validation has been published with regard to glacier melt (besides a general validation of this model suite for a different setup in Avanzi et al. 2022a).

For glacier-mass balance data to validate melt estimates, we employed a dataset of 32 ablations stakes in Aosta Valley and 39 in Lombardy, installed on the various glaciers shown in Figure 1 at elevations from 2546 m to 3545 m a.s.l.. Altogether, 208 measurements were collected from 2009 to 2022 by the Environmental Protection Agency of Aosta Valley and the Lombardy

Glaciological Service, the two institutions responsible for glacier data collection in Aosta Valley and Lombardy, respectively (the latter in collaboration with the Regional Environmental Protection Agency of Lombardy and the Snow and Meteorology Monitoring Center of Bormio). Despite the inherent uncertainty and spatial limitations of ablation stake data (Cuffey and Paterson, 2006; Fountain and Vecchia, 1999), comparing modeled glacier melt to these measurements represents the most informative validation of S3M glacial melt component feasible to date.

Streamflow data for the Dora Baltea and Adda rivers were provided by the Autonomous Region of Aosta Valley and the Environmental Protection Agency of Lombardy, respectively. The dataset consisted of water-stage measurements, which were





converted into flow rates using flow rating curves provided by the two respective institutions and accounting for morphological changes. For both datasets, we removed unreliable measurements to ensure data quality, mainly through visual screening. Both basins have a high degree of anthropization, especially due to hydropower, which may alter the timing and magnitude of river flow. As we will further discuss in Subsection 5.1, however, several pieces of evidence show that this alteration was minor for our scopes and at our scales.

In this paper, we will always refer to water years, defined as periods of time between September 1 and August 31.

## 3.2  Snow drought characterization

We characterized the snow drought events of 2022 and 2023 by using daily maps of air temperature, total precipitation, and snow water equivalent (SWE) as available from S3M Italy across our study area (Avanzi et al., 2023). These data were processed to extrapolate daily trajectories of air temperature, precipitation, and mean snow water equivalent across select elevation bands over glaciers (elevation bands were 2000–2500 m, 2500–3000 m, 3000–3500 m, 3500–4000 m, and 4000–4500 m a.s.l.).

To highlight the deviations from typical conditions, we first computed daily quartiles of air temperature, total precipitation, and snow water equivalent over the period 2010-2023, and then plotted these quartiles along with the trajectories of daily air temperature, total precipitation, and snow water equivalent for 2022 and 2023. Air temperature and snow water equivalent data were smoothed using a 10-day moving average to better highlight key events during these two years at different elevations. Precipitation data were cumulated by elevation band.

We also calculated seasonal anomalies for air temperature and precipitation in 2022 and 2023 relative to the mean seasonal values from 2010 to 2023. Each season corresponded to a three-month period according to meteorological conventions for this region: winter (December, January, February), spring (March, April, May), summer (June, July, August), and autumn (September, October, November).

## 3.3  Quantification of the glacier melt and validation strategy

To validate S3M Italy for the glacier component, we compared point mass balance data from 32 ablation stakes with co-located glacier melt model estimates. Since each ablation-stake data reports the mass balance over a certain period of time (usually, from summer to summer), for each data point we accumulated glacier melt for the same location and the same period of time. Subsequently, we computed correlation index, bias (mean difference between simulated and observed values), a confusion matrix, the Root Mean Squared Error, and the coefficient of determination between observed and modeled cumulative glacier melt.

A rigorous quantification of glacier melt contribution would require knowing the proportion of total discharge that ids directly attributable to glaciers, and thus a complex glacio-hydrologic model incorporating evapotranspiration and groundwater recharge. Lacking such a model, we used a simplified approach, estimating this contribution as the ratio of cumulative glacier melt to total observed discharge over a given period (Gascoin, 2024). We converted cumulative glacier melt in mm into a flow rate with units matching streamflow ($m^3$/s), by multiplying glacier melt by the pixel area (40000 $m^2$). Then, we computed weekly ratios between cumulative, basin wide glacier melt and cumulative streamflow as a proxy for the contribution of glacier





melt to streamflow. We chose a weekly resolution for our analysis in order to account for the time needed by glacier melt to reasonably influence streamflow at the closure sections. Since no definitive estimate of this concentration time is available for these regions, this temporal resolution should be seen as a trade-off between shorter (such as daily) or longer (such as monthly) resolutions, which would both be inevitably too short or too long to capture driving factors of the glacier-streamflow interaction. We also evaluated the relationship between annual and daily glacier melt and streamflow to gain further insights.

## 4 Results

### 4.1 S3M Italy model validation

Despite the significant spatial mismatch between stake measurements and a 200x200 m$^2$ model, melt-estimate biases were generally within a $\pm 1$ m w.e. range (Figure2). The model performed best between 2800 m and 3200 m a.s.l., which is the most critical zone, as a significant portion of glacier mass currently lies at these elevations in the Italian Alps (Paul et al., 2020).

At lower elevations, the model generally underestimated melt, likely due to the model struggling to represent the presence of debris on the glacier surface or other tongue processes. From 3300 m a.s.l. upward, instead, the model tended again to underestimate melt, possibly due to an overestimation of snow accumulation by the model or an underestimation of glacier melt at these elevations. These underestimations will result in a conservative estimate of glacier contribution to streamflow.




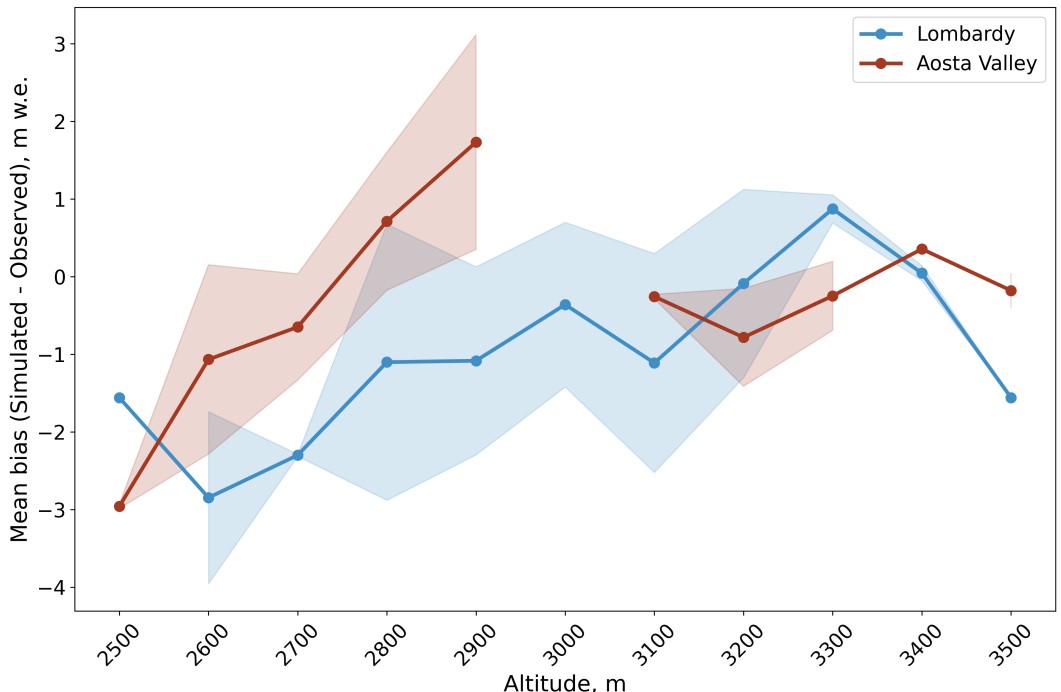

**Figure 2.** Mean bias of S3M (modeled values) against stake measurements (observed values), binned by elevation, for both Aosta Valley (in red) and Lombardy (in blue). Shaded areas show the standard deviation (sd). Missing data points for Aosta Valley are because data were unavailable between 2900-3000 and 3000-3100 m a.s.l.

## 4.2 Air temperature and precipitation over glaciers

In Aosta Valley, both 2022 and 2023 saw significant warming, especially during winter and summer (Figure 3). During the winter quarter (December 2021, January - February 2022), for example, mean temperature was -2,82 °C compared to a mean of -4,66 °C at 2000-2500 m. Similarly, at 2500-3000 m, it was -5,22 °C (vs. -7,11 °C); at 3000-3500 m, -6,25 °C (vs. -8,48 °C); at 3500-4000 m, -7,33 °C (vs. -10,07 °C); and at 4000-4500 m a.s.l., -8,68 °C (vs. -11,71 °C). May 2022 saw a significant heatwave, and then June and July temperatures were again notably warmer than the median (Figure 3). Specifically, at 2000-

2500 m a.s.l., the mean reached 11,05 °C, significantly above the 9,38 °C average. Further, temperatures at 2500-3000 m were 8,17 °C (vs. 6,41 °C); at 3000-3500 m, -6,85 °C (vs. 4,70 °C); at 3500-4000 m, 4,97 °C (vs. 2,45 °C); and at 4000-4500 m a.s.l., 2,73 °C (vs. 0,16 °C). Compared to 2022, the anomalies in both winter and summer of 2023 were more reduced (see again Figure 3). Results were somewhat similar in Lombardy (Figure A1), despite a colder-than-average winter in 2022 (exemplified by -6,63 °C at 2000-2500 m vs. -5,83 °C on average) and then consistently higher-than-usual summer temperatures.

Warmer-than-average conditions during summer were driven by different mechanisms in 2022 vs. 2023. In 2022, warm spells during late spring and early summer played a key role, with distinct periods of high temperatures observed in May, June, and July (see again Figure 3). For instance, May 2022 recorded an average of 25 days exceeding the daily mean temperature of



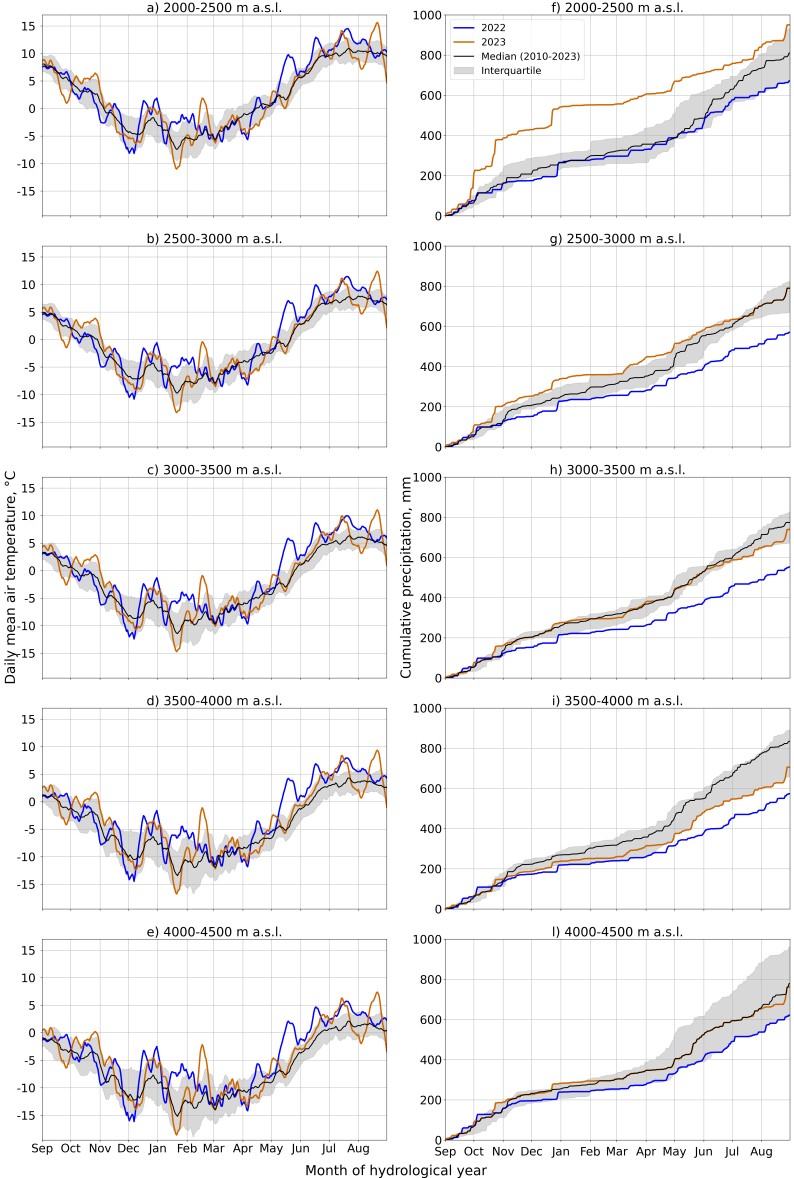

**Figure 3.** Daily mean air temperature (a to e) and cumulative precipitation (f to l) by elevation band over the glaciers of Aosta Valley during 2022 (in blue) and 2023 (in orange line) compared to the median 2010-2023 (in black) and the interquartile range (grey area).

the 2010-2023 period, with 17 of these days showing a difference of +2 °C or more. Similarly, June experienced 24 days with temperatures above average (of which 16 days were at least 2 °C above normal), with the peak of this warm spell occurring
around June 18. July recorded its highest temperatures on July 18th across all elevation ranges, with 27 days above the 2010-2023 average and 19 days with temperature exceeding 2 °C the period of reference. 2023, on the other hand, saw later and more



concentrated hot spells, coupled with a cooler spring. Two main periods of above-average temperature concentrated in July and August 2023, particularly from July 8 to 20, with 22 days above the 2010-2023 average, 12 of which were more than 2 °C above normal at all altitudes, and then another from August 10 to 25, averaging 16 days more than 2 °C above the long-term
mean. Results were similar in Lombardy (Figure A1), with 2022 characterized by distinct warm spells in late spring and early summer (e.g., May 11 - May 28), while 2023 experienced later and more concentrated hot spells, particularly in August (e.g., Aug. 10 - Aug. 27).

In terms of precipitation (Figure 3, right column), Aosta Valley experienced a drier-than-average 2022, particularly in winter and spring, followed by a generally wetter 2023 at lower elevations and more in line with the median at higher elevations. For
instance, winter precipitation at 3500-4000 m in 2022 was 66,8 mm, a significant deficit compared to the 2010-2023 average of 88,4 mm. Spring at 3000-3500 m also showed a notable deficit in 2022, with 132,2 mm vs. the average of 196,9 mm. Across all elevations, September and October precipitation was close to the average in 2022, but a clear deficit became evident again from November onward. At the highest elevation range (4000-4500 m a.s.l.), the overall cumulative precipitation was closer to the median, generally remaining within the interquartile range (e.g., Autumn: 195,3 mm vs. average 204,4 mm). 2023
showed a distinctly different pattern, with the elevation ranges between 2000 and 3000 m a.s.l. being particularly wetter than average. For example, at 2000-2500 m a.s.l. cumulative precipitation largely exceeded the average, notably in autumn with 429,1 mm compared to the 2010-2023 average of 236,4 mm (note that this deviation was due to 2-3 specific storms, partially of convective nature). At 2500-3000 m a.s.l., spring saw 219,4 mm versus an average of 192,3 mm, and precipitation stayed above average from October 2022 through June 2023. For higher elevations (3000 m to 4500 m a.s.l.), cumulative precipitation
was largely on average. Results were similar in Lombardy (Figure A1): 2022 was generally drier-than-average across most seasons and elevations, notably in winter with cumulative precipitation at 2000-2500 m a.s.l. drastically reduced to 44,9 mm (compared to 128,1 mm average). Conversely, 2023 saw generally higher cumulative precipitation, particularly in summer, where 2000-2500 m a.s.l. recorded 519,2 mm (vs. 383 mm average), though winter remained significantly drier-than-average, similar to 2022.

Seasonal temperature anomalies showed a clear signature of elevation-dependent warming across both years and all seasons in Aosta valley (Figure 4). Winter and summer experienced the highest temperature anomalies (winter: +1,84 °C to +3,02 °C; summer: +1,67 °C to +2,57 °C), making them the warmest seasons of 2022. Spring also showed positive anomalies, which increased significantly with elevation. Concerning 2023, the only clear difference compared to 2022 was that spring was notably colder than other seasons (ranging from -0,62°C to +0,68 °C), which led to some late-spring snowfalls. These
results were similar in Lombardy (Figure A2), where a recurring pattern emerged across most seasons: a negative anomaly at the lowest elevation band (e.g., -0,35 °C in summer 2022 at 2000-2500 m a.s.l.), followed by increasing positive anomalies with altitude up to 3000-3500 m e.g., +2,09 °C at 3000-3500 m a.s.l. in summer 2022), and then a slight decrease in the positive anomaly at the highest band (3500-4000 m a.s.l.); with spring 2023 notably colder at lower elevations (e.g., -1,64 °C at 2000-2500 m a.s.l.).





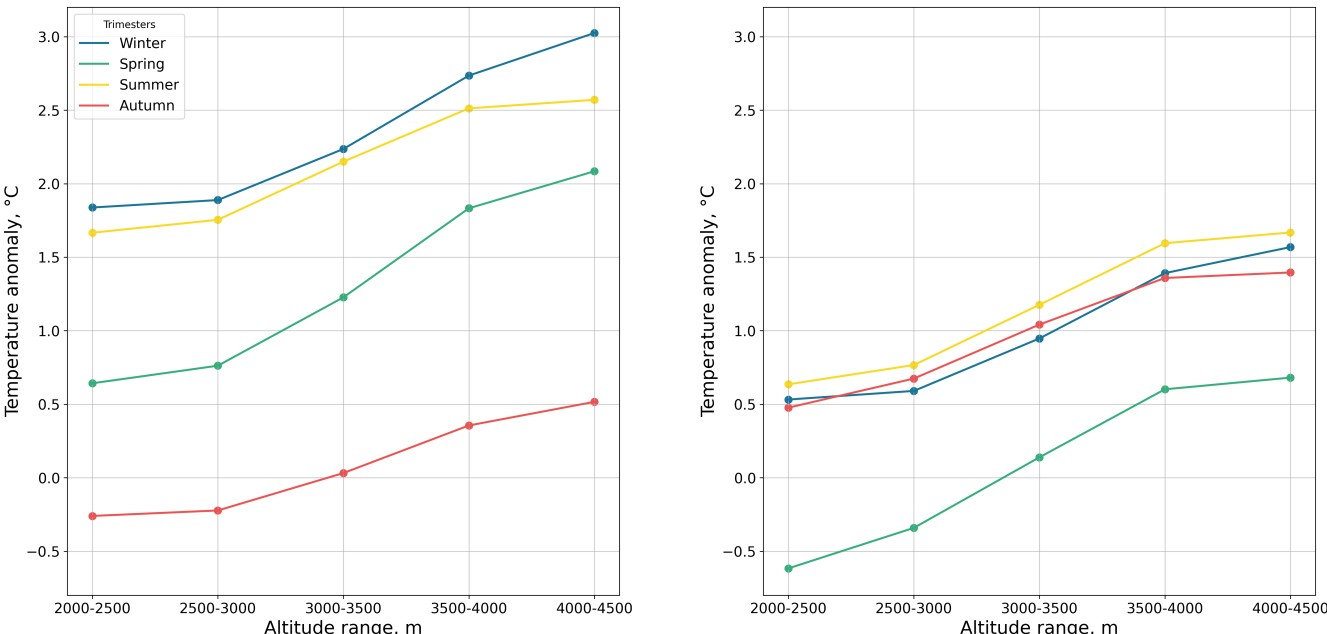

**Figure 4.** Seasonal air temperature anomalies as a function of elevation bands in Aosta Valley for the years 2022 (left) and 2023 (right).

## 4.3 Snow water equivalent over glaciers

Both 2022 and 2023 saw substantial and widespread snow water equivalent (SWE) deficits over glaciers across all elevations in Aosta Valley (Figure 5). In 2022, the snow water equivalent anomaly reached approximately -76 % across the 2000–3500 m a.s.l. range. This deficit lessened slightly at higher elevations, recording -64 % for 3500–4000 m a.s.l. and -47 % for 4000–4500 m a.s.l.. Similarly, 2023 exhibited substantial snow water equivalent deficits, though with slightly different altitudinal patterns. The anomaly was approximately -75 % from 2000–3500 m a.s.l., -71 % at 3500–4000 m a.s.l., and -64 % at 4000–4500 m a.s.l..

While this significant snow water equivalent deficit persisted in both 2022 and 2023, the timing of snow accumulation and melt differed notably: 2022 saw an early end to the snow accumulation season at all elevations (up to two months), which was directly attributable to the combined effect of consistently higher temperatures and lower precipitation. This premature end of accumulation inevitably led to an earlier onset of the melt season, which reduced the duration of snow cover on glaciers (melt out dates in 2022 were approximately 1,5 months earlier than usual at 2000-2500 m a.s.l.) Conversely, in 2023, the lower temperatures during spring, coupled with late spring snowfalls as visible in Figure 5, extended the snow accumulation season closer to its average end date.

Similar to the patterns observed in the Aosta Valley, Lombardy also faced substantial and widespread snow water equivalent (SWE) deficits across all analyzed elevations in both 2022 and 2023 (Figure A3), clearly highlighting severe snow drought conditions. In 2022, the snow water equivalent mean anomaly reached -63,2 % at 2000-2500 m a.s.l., intensifying to -69,2 %





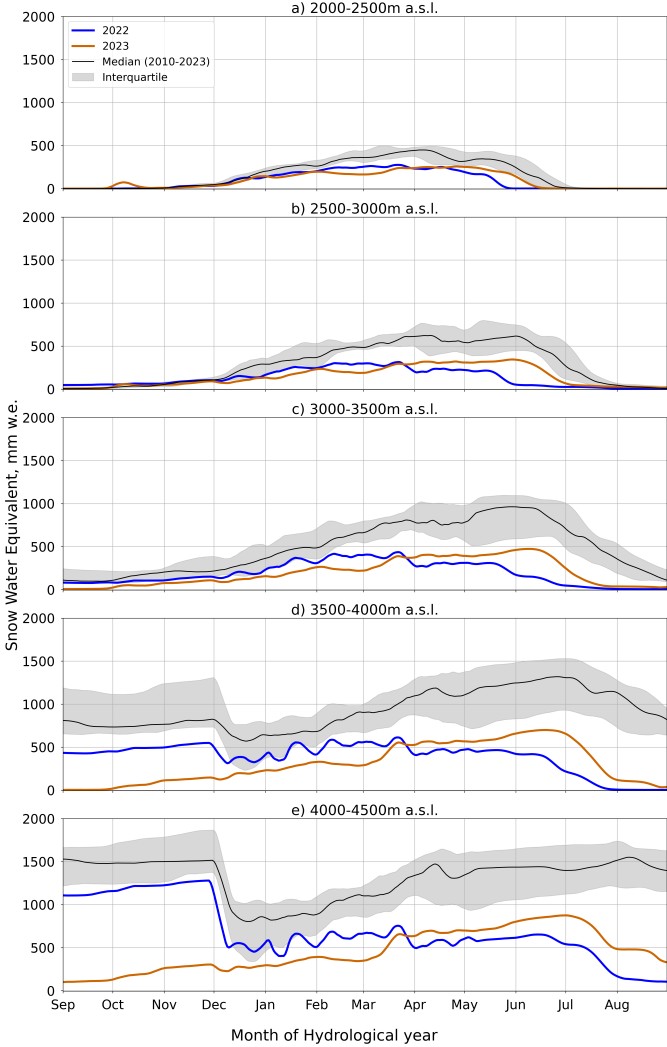

**Figure 5.** Daily snow water equivalent (SWE) by elevation range over the glaciers of Aosta Valley during 2022 (in blue) and 2023 (in orange line) compared to the median 2010-2023 (in black) and the interquartile range (gray area).

for 2500-3000 m a.s.l., and then settling at -63,0 % for 3000-3500 m a.s.l.. This deficit decreased significantly at the highest elevations, recording -37 % at 3500-4000 m a.s.l.. Similarly, 2023 exhibited substantial snow water equivalent deficits, though with slightly different altitudinal patterns. The anomaly was approximately -70,9 % at 2000-2500 m a.s.l., -65,2 % at 2500-3000 m a.s.l., and -55,6 % at 3000-3500 m a.s.l.. The deficit was again notably lower at the highest analyzed band, reaching -25,9 % at 3500-4000 m a.s.l..




## 4.4 Glacier melt contribution to streamflow

The 2022 and 2023 droughts markedly altered hydrology in both catchments, with extremely low river flows highlighting an intense summer hydrologic drought following the winter snow drought (Figure 6): annual mean streamflow in 2022 and 2023 was approximately -32 % lower than the mean annual streamflow between 2011 and 2021, contrasting with a higher-than-usual glacier melt ($\sim + 148$ % mean annual glacier melt compared to 2011-2021).

More in details, a clear contrast was observed during the 2022 water year: average annual streamflow was among the lowest recorded during the study period (64,5 m³/s in Tavagnasco and 60 m³/s in Fuentes), while annual mean glacier melt was the highest over the period of record (9,5 m³/s in Fuentes and 14,93 m³/s in Tavagnasco). In 2023, instead, mean annual streamflow and glacier melt rates were slightly lower than the previous year at both Tavagnasco and Fuentes. At Tavagnasco, the mean streamflow measured 69,9 m³/s, with a glacier melt of 10,2 m³/s. Meanwhile, at Fuentes, the mean streamflow was 61 m³/s, and glacier melt averaged 8,27 m³/s.

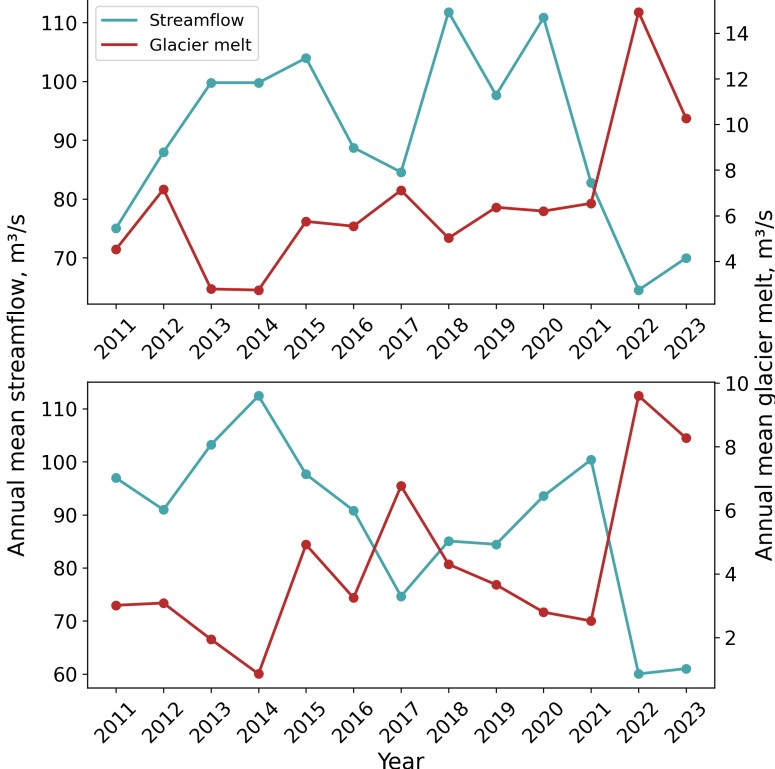

**Figure 6.** Comparison between annual mean streamflow (in blue) and annual mean glacier melt (in red, simulated), Dora Baltea (top) and Adda (bottom) rivers, water years 2011 through 2023.

Glacier melt contribution to streamflow during summer doubled, or nearly tripled, in both catchments during 2022 and 2023 compared to pre-2022 water years (Figure 7). This spur in glacier melt and concurrent decline in streamflow highlighted a clear





signature of snow droughts on the link between glacier melt and water supply - a signature that resolves around four significant

mechanisms (Figure 7).

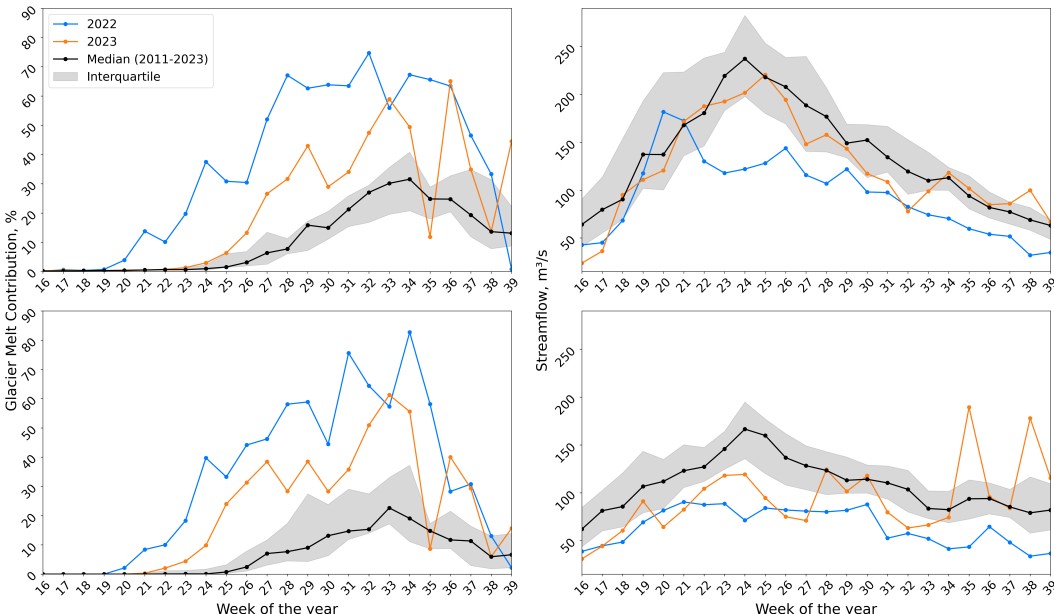

**Figure 7.** Weekly glacier melt contribution to streamflow (left) and weekly mean streamflow (right) for water year 2022 (in blue) and water year 2023 (in orange) compared to the median and the interquartile range for water years 2011 - 2023, Tavagnasco (up) and Fuentes (bottom). Note that we included water years 2022 and 2023 in the interquartile range due to the comparatively short period of record.

The first mechanism is an earlier-than-usual start of the glacier-melt season (defined here as the week when glacier contribution first exceeded 5 %): in both Aosta Valley and Lombardy the 2022 melt season began six weeks earlier than the median for 2011-2021. This earlier-than-usual start took place also in 2023: two weeks ahead of the median in Aosta Valley and three

weeks ahead in Lombardy.

The second mechanism is an increase in the contribution of glaciers to streamflow during the whole of the melt season, and not just the peak summer-melt period. In 2022, peak glacier-melt contribution to streamflow reached 75 % in Tavagnasco, while it reached 65 % in 2023, two to three times the usual contribution for this region (31,5 %). At Fuentes, the impact was even more pronounced: in 2022, peak contribution nearly quadrupled (83%), and in 2023, it almost tripled (61 %) compared

to the median values (22,6 %).

The third mechanism is a potential shift in the timing of the seasonal glacier melt peak, which was particularly evident in Tavagnasco in 2022, when summer rainfall greatly decreased (Avanzi et al., 2024) and thus the glacier melt peak stood out clearly. During this year, peak contribution occurred two weeks earlier than usual (from week 34, Aug. 22 to Aug. 28, to week 32, Aug. 8 to Aug. 14). At Fuentes, the glacier melt peak during 2022 was recorded during week 34, that is, closer to the





median (week 33, Aug. 15 to Aug. 21), although a first sub-peak was already observed during week 31 (Aug. 1 to Aug. 7). In 2023, on the other hand, sporadic rainfall during summer hindered the assessment of the glacier melt peak.

The fourth and final mechanism observed is a potential prolongation of the melt season, which underpinned a higher-than-usual glacier melt contribution than the median even during late summer / early autumn. In Tavagnasco, glacier melt in 2022 accounted for 33 % of the streamflow even during week 38 (Sep. 19 to Sep. 25), significantly above the median of 13 %.

Similarly, at Fuentes, glacier melt during week 37 (Sep. 12, to Sep. 18) reached 31 %, compared to the median of 11 %. This effect was even more pronounced in 2023 due to higher late-summer temperatures, as shown in Figure 8. During 2023, in Tavagnasco, glacier melt during week 39 (Sep. 26 to Oct. 2) contributed 44 % of streamflow, well above the median of 13 %, while in Fuentes, the contribution during week 39 was 15 %, compared to the median of 7 %.

Despite these four general mechanisms, glacier melt contribution to streamflow remains sensitive to short-term meteoro-

logical events, such as temperature drops and early or late snowfalls (Figure 8). For instance, between April and May 2023, streamflow quickly increased in Aosta valley due to the seasonal freshet, which continued until June. Then, streamflow decreased slightly as the seasonal snowfall waned. In June, glacier melt increased, quickly becoming a dominant contributor to streamflow. Glacier melt peaked in late August, coinciding with the peak of the glacier melt season. A sudden drop of temperatures between late August and early September, however, led to a quick and sudden glacier melt decrease, and to a pairwise

decline in streamflow (Figure 8, a and b panel). In early September, finally, glacier melt rapidly rose as temperature increased again, growing at 63 m$^3$/s in Tavagnasco and 42,3 m$^3$/s in Fuentes.

## 5   Discussion

This study provides comparatively new insights into the crucial role played by glacier meltwater in sustaining and modulating streamflow during droughts. These insights come as the enumeration of four signature mechanisms caused by the impact of a

snow drought on a heavily glacierized system, namely an early onset of the glacier melt season, an intensification of glacier melt contribution, an earlier seasonal peak in glacier melt contribution, and an extension of the glacier melt season.

Regarding the earlier-than-usual onset of the melt season (mechanism 1), previous research by Thibert et al. (2018) has already established a long-term trend towards an earlier onset of the melt season, consistent with our findings. For example, Farinotti et al. (2012) found that future runoff in the Swiss Alps will be characterized by a significant shift towards earlier melt

runoff, progressively advancing with each decade. Our study further substantiates this trend, showing how snow droughts are a pivotal driver in such long-term trends and a potential harbinger of conditions to come in a warmer and drier world. In such future conditions with less snow on the ground, glaciers will be even more central in the mitigation of hydrologic droughts (Van Tiel et al., 2021).

The intensification in glacier melt contribution to streamflow (mechanism 2) also aligns with previous studies(Huss and

Hock, 2018; Farinotti et al., 2012), which have shown that this intensification takes place particularly during droughts (Pellicciotti et al., 2010). However, results presented in Section 4.4 reveal not only an intensification during the early stages of the melt season, but a sustained increase throughout the entire melt season. In both regions, the 2022 snow drought led to a more



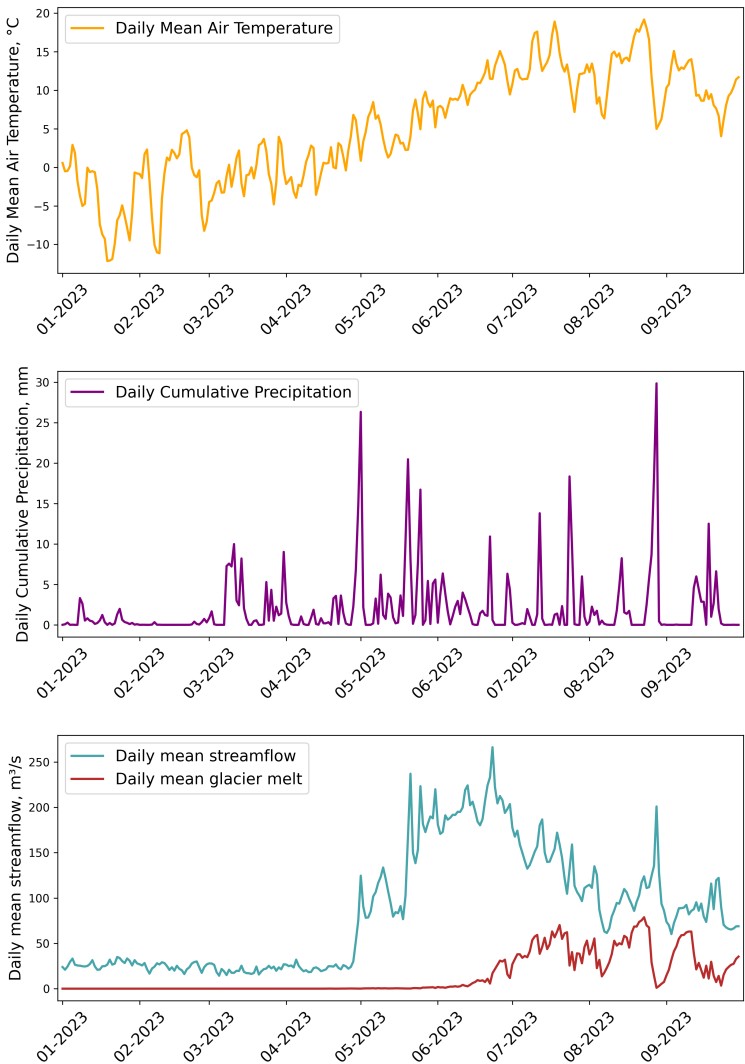

**Figure 8.** Daily mean air temperature (a), cumulative precipitation (b) across Aosta Valley, and daily streamflow and glacier melt for water year 2023 in Tavagnasco (c).

intense glacier melt season compared to pre-drought years, with glacier contributions to streamflow remaining significantly enhanced throughout the entire season, also doubling or tripling the average peak values (Figure 7). Water year 2023 proved

that, despite its greater meteorological variability during summer, the glacier melt contribution kept at a significant level during the whole season. This second mechanism means that, in glacierized regions, glacier melt sustains streamflow throughout the season, rather than only during the peak-melt period, and that this holds particularly during snow droughts and periods of warm, dry weather (Cremona et al., 2023).





Regarding the potential shift in the seasonal peak-melt timing (mechanism 3), Farinotti et al. (2012) predicted a dramatic
shift in runoff patterns in the Swiss Alps due to climate change, with both the onset and peak of the melt season occurring
progressively earlier. Such a shift was clearly observed in our study region for 2022 in Tavagnasco, while results for 2023
and in general at Fuentes are less conclusive in this regard. We explain this as follows: during 2022, the winter snow drought
immediately left room to a significantly dry and warm summer with little to none residual snow on glaciers (Avanzi et al.,
2024). Under such conditions, glacier melt was likely strongly radiation-dominated, with a peak in July. During 2023, instead,
some late-season snowfall provided some marginal snowpack over glaciers in the Italian Alps, which likely reduced radiation-
driven melt and thus re-aligned peak-melt timing to 2011-2021. While this analysis would clearly benefit from more data, it
nonetheless points to the often overlooked, but likely important role of marginal snowpack and out-of-season snowfalls on
glacier preservation (Fyffe et al., 2021).

The fourth observed mechanism is the lengthening of the melt season. The 2023 data presented in Figure 7 shows increased
glacier melt in September and October of 2023 at both Tavagnasco and Fuentes, highlighting the growing importance of
late-season melt in compensating for reduced snow cover, low precipitation, and high temperatures (Cremona et al., 2023).
Furthermore, abrupt transitions between temperature drops and high melt rates (Figure 8), observed twice between September
and October (particularly in the first week of September – weeks 35 and 36), showcase the influence of compound drought
and heatwave events on cascading cryosphere-hydrologic droughts. Weeks 36 to 39 (excluding seasonal variability) clearly
demonstrate the impact of such events on extending the melt season in 2023. This outcome further emphasizes the crucial role
of late-season glacier melt in sustaining streamflow and the potential for snow droughts to disrupt typical glacier melt patterns.

## 5.1 Assumption and limitations

This study employed S3M Italy for glacier melt estimates. In its current operational setting, this model does not account for
glacier movement and debris cover on glaciers (Avanzi et al., 2022a). As already pointed out, the impact of glacier movement
on our findings is likely a second-order problem, given that we considered a 13-year period (Huss et al., 2010; Bongio et al.,
2016). On the other hand, thick debris typically acts as a protective layer for the underlying ice (Fyffe et al., 2019), which may
have led to a local overestimation of glacier melt at very low elevations (note that these elevations are outside the validation
range we considered in Figure 2). To account for both aspects, future research with S3M should explore including spatially
distributed estimates of debris cover such as that by Rounce et al. (2021) and an explicit simulation of glacier movement.

S3M Italy employs static glacier maps from the Randolph Glacier Inventory v 6.0 (Avanzi et al., 2023), which is a globally
complete inventory intended to capture the world's glacier outlines near the beginning of the 21st century (RGI Consortium,
2017). Consequently, these static glacier outlines may overestimate current glacier extent, particularly during the drought years
taken into consideration. To assess this discrepancy, we leveraged more recent glacier outlines provided by the Aosta Valley
Autonomous Region (2019 data) and the Lombardy Glaciological Service (SGL) (2021 data), comparing them against the
Randolph Glacier Inventory v 6.0 used by S3M Italy. Our analysis revealed a glacier area loss of 6,97% in Aosta Valley
between 2000 and 2019. Lombardy experienced a 18,66% reduction in glacier coverage from 2000 to 2021. However, the
glacier outlines in the RGI dataset better represent glacier extents at the beginning of our simulation period (2010) than at the





end (2022–2023). As glaciers have been retreating over the study period, the fixed geometry used from RGI overestimates the glacierized area contributing meltwater in recent years. Even under the simplistic assumption that glacier melt contributions to
streamflow declined linearly with glacier area, this would suggest that melt contributions in 2022 and 2023 were still at least double the median contributions from the earlier part of the period (2011–2023), which validates our results.

Another potential limitation of this study is the presence of artificial reservoirs in both the Adda and Dora Baltea basins, which may have impacted the natural timing of water transit. While reservoir impacts depend on energy market fluctuations (Guo et al., 2021), we in general expect hydropower to shift runoff from early to late summer, as spring freshet is accumulated
to meet later peaks in energy prices. With regard to our study region, a recent study by Amaranto et al. (2023) has reconstructed the naturalized streamflow of the Adda basin, specifically as inflow to Lake Como, thus the section that we are interested in, and demonstrated that the resulting effect of high-elevation reservoirs is of relatively limited magnitude for such valley streamflow gauges (that is, daily differences were on the order of 10 % when considering the climatology of streamflow at Fuentes). The largest discrepancies between full-natural and observed streamflow occur during spring and early summer, a period when
snow melt contributions are most substantial. By July and August, these differences decrease markedly and remain minimal throughout the remainder of summer and fall. While similar estimates are missing for Aosta Valley, we expect these dynamics to be similar in such proximal contexts.

The importance of glaciers in sustaining summer water supply during droughts regardless of reservoir operations is also clear if once looks at annual rather than weekly time scales. In fact, the mean annual glacier melt contribution in Aosta Valley
doubled from 6 % (2011-2021) to 18 % (2022-2023). In Lombardy, the increase was even more pronounced, with glacier melt contribution nearly quadrupling from 4 % (2011-2021) to 15 % (2022-2023).

## 6  Conclusions

This study examined the 2022 and 2023 snow droughts in the Italian Alps and found that significantly increased glacier melt (up to +148 % compared to 2011-2021) and low streamflow led to glacier melt contribution to summer streamflow in the
Aosta Valley and Lombardy regions reaching peak levels of 75 % and 83 % in 2022 and 65 % and 61 % in 2023, respectively, highlighting the crucial role of glacier melt in mitigating snow drought impacts. Four key mechanisms were identified: an earlier melt season onset (up to six weeks compared to the 2011-2021 median), intensified glacier melt contributions, a potential shift in peak melt timing, and a prolonged melt season into late autumn, all demonstrating the increased importance of glaciers in sustaining streamflow during severe droughts and emphasizing the vulnerability of alpine water resources.

*Data availability.* Sources of data used in this paper are reported in Sect. 3 and are derived from the Aosta Valley Regional Authority (https://cf.regione.vda.it/it/, last access: 18 November 2024), the Aosta Valley Environmental Protection Agency (https://www.arpa.vda.it/, last access: 18 November 2024), the Lombardy Environmental Protection Agency (https://www.arpalombardia.it/, last access: 18 November 2024), and the Lombardy Glaciological Service (www.servizioglaciologicolombardo.it). These data were made available by such third parties, which retain copyright. Outputs by S3M Italy are available at https://doi.org/10.5281/zenodo.6861722.





**Appendix A: Appendix A**



**Figure A1.** Daily mean air temperature (a to d) and cumulative precipitation (e to h) by elevation range over the glaciers of Lombardy during 2022 (in blue) and 2023 (in orange line) compared to the median 2010-2023 (in black) and the interquartile range (gray area).



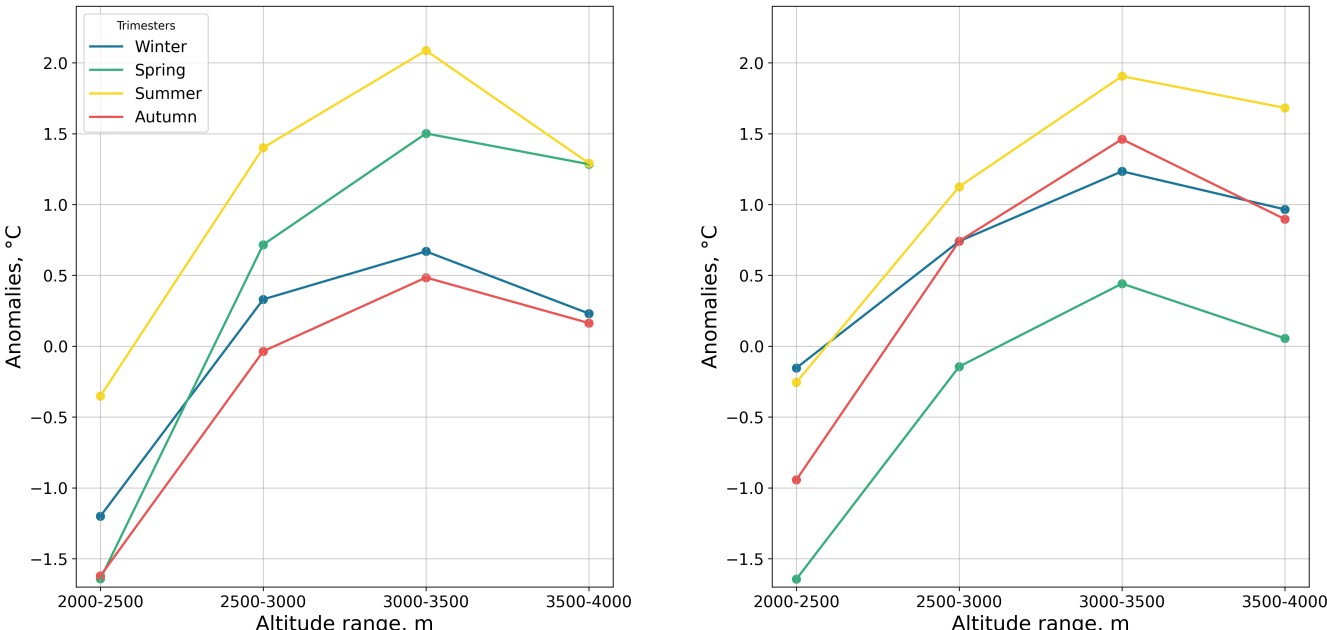

**Figure A2.** Seasonal air temperature anomalies in function of elevation ranges in Lombardy for the years 2022 (left) and 2023 (right).



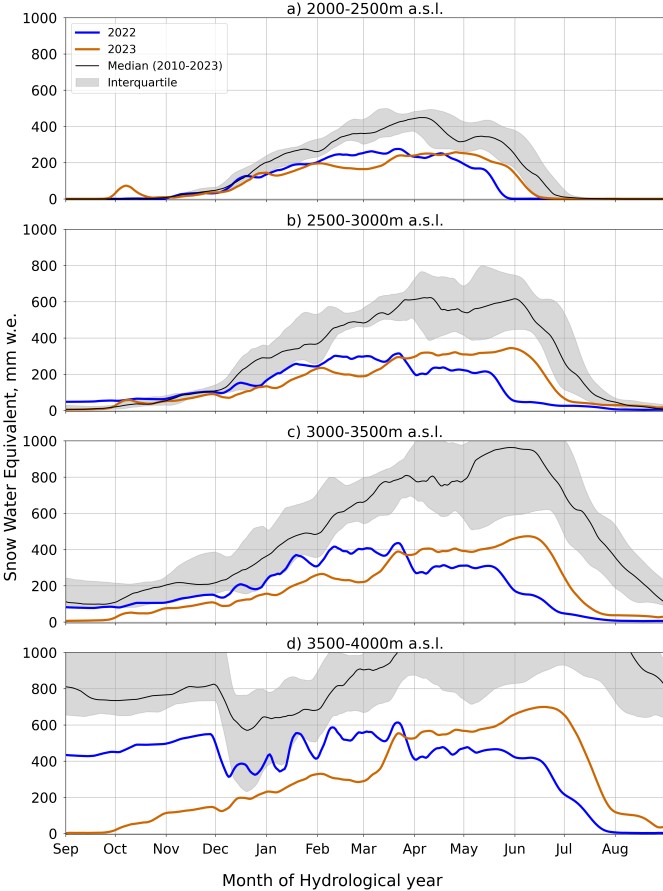

**Figure A3.** Daily snow water equivalent (SWE) by elevation range over the glaciers of Lombardy during 2022 (in blue) and 2023 (in orange line) compared to the median 2010-2023 (in black) and the interquartile range (gray area).



*Author contributions.* M.L. designed the study, processed the data, performed the analyses, and wrote the manuscript. F.A. designed the study and contributed to the modeling strategy, data interpretation, and manuscript revision. U.M.D.C., E.C., M.I., and P.P. provided cryospheric data and regional expertise for Aosta Valley. R.S. and A.M. contributed with glacier data for Lombardy and regional expertise for Lombardy. L.F. and R.C. supervised the research and contributed to manuscript refinement. All authors reviewed and approved the final version of the
manuscript.

*Competing interests.* The authors declare that they have no competing interests.

*Acknowledgements.* This research was partially funded by the European Union - NextGenerationEU and by the Ministry of University and Research (MUR), National Recovery and Resilience Plan (NRRP), Mission 4, Component 2, Investment 1.3 "The creation of extended partnerships with universities, research centers, and companies for the funding of basic research projects" PE00000005 "Multi-Risk sciEnce
for resilienT commUnities undeR a changiNg climate (RETURN)" CUP B57G22001180002.



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
