# Peer review of "The 2022-2023 snow drought in the Italian Alps doubled glacier contribution to summer streamflow"

_EGUsphere, 2025_

## Referee Comment (RC2)

**Review of Leone et al. The 2022-2023 snow drought in the Italian Alps doubled glacier contribution to summer streamflow**

**Overall comments**

Leone et al. use the operational model S3M, combined with discharge measurements to estimate the glacier contribution to discharge during the 2022-2023 snow drought in the Italian Alps. They find large glacier melt contributions to discharge during this period and explain that these high contributions were a result of a combination of an extension of the glacier melt season (starting earlier and ending later), increased melt rates and an earlier peak in glacier melt.

Overall, I think it is an important piece of work since the extreme conditions in these years and the particular role of glacier runoff under such conditions warrants study, especially given that the drought conditions, combined with increased temperatures found in these years might be similar to future conditions. It was also useful that the authors were able to determine the particular effects of these extreme conditions on the glacier melt contribution. However, I do think the paper would benefit from some improvements prior to publication.

a) **Description of results.**
   In some places I find the description of results to be a little too detailed, in terms of including many statistics in the main text which could be easily summarized more succinctly or the reader could be referred to a figure. Sometimes just shortening the text is enough, whereas in other cases improving the current figures or adding new ones would be beneficial. I think this would help keep the story clear for readers.

b) **Clearer description of (some) of the study limitations as well as model uncertainty.**
   I think there could be an improvement in the description and illustration of the comparison of the model results against glacier mass balance (see detailed comments), and potentially remote sensing datasets could be used for comparison too. It would also be useful (if possible) to quantify the effect of not evolving the glacier outlines. The authors are clear about how they estimate the glacier contribution to discharge and already mention that it is a simplification, but I think it likely leads to an overestimation of glacier contribution to discharge since many of the not included elements (groundwater and ET) result in losses and therefore would increase the calculated glacier contribution. Potentially the glacier melt as a proportion of inputs (rain, snowmelt and ice melt within the catchment) could be used as an alternative and/or other studies could be used to compare against the magnitude of the results (even if they are in a slightly different region of the Alps). This could go in the discussion and would allow your results to be seen better in context.

c) **Treatment of snowmelt**
   It seems like the SWE (at least over glaciers) is calculated for the catchments and that the glacier melt contribution is a combination of the snow and ice melt over the glaciers, which is pretty reasonable. However, it is less clear to me how off-glacier snow is dealt with. In section 3.1 it is mentioned that SWE is calculated by S3M nationally (so I presume also off-glacier), and yet the snow contribution to discharge is not discussed later in the paper, only the on-glacier SWE variations. Maybe for this study off-glacier snow melt was not calculated? I respect that the authors may have chosen to concentrate specifically on the glacier melt contribution, but I think

it either needs to be said clearly why the off-glacier snow melt component was not calculated/analysed or for it to be included alongside the glacier melt contribution.

d) **Figures**

I think in various places the Figures could be improved (please see comments below) and the Appendix figures (which are for Lombardy) could fit in the same figures as the Aosta ones which are in the main paper.

**Minor comments**

Abstract: consider starting with an introduction/rationale sentence

Line 10: maybe 'elevation-dependent' is more correct

L11: Be careful to use points 1.5 (rather than 1,5) here, and throughout the manuscript

L15: early/late season snowfalls

L24: 'negative glacier mass balances' – maybe the argument isn't quite the right way round here, since the climate warming causes the negative glacier mass balances, which are driven by the decreased snowfall

L35 'dry and warm conditions' do you really mean both dry and warm conditions, for snow droughts are you defining them as specifically a lack of precipitation? Or are the warm temperatures also important?

L37 and 38: I agree that the full process from glacier melt to water supply deficit are not fully understood, but there are well known links between low snow and its impact on glacier melt and mass balance.

L43: 'the contribution of glacier melt to streamflow' and 'early studies' – I would tend to say previous studies, as 'early' tends to indicate quite old studies whereas you have included those which are fairly recent

L46: 'whereas elucidating the' might be better here

L48: 'little to no contribution'

L49: Ayala et al is a Chilean study, it is relevant to your work, but here you are talking about Mediterranean climates. Also 'a crucial link' might be better phrasing than 'crucial hinge'.

L51: remove extra brackets around citation

L58: Missing a year for the citation, here and also on line 63

L65: '2022-2023 season' Just for clarity is this one hydrological year, or just the melt season? Maybe give dates

L66: Maybe rephrase for clarity, ''Both 2022 and 2023 were characterised by intense heat and exceptionally low snowfall accumulation, demonstrating..'

L71: Would it be correct to say '2022 and 2023 winter snow drought'? If so this is perhaps clearer.

L71 to 73: Maybe stay in the same tense throughout this paragraph, this is just a style thing to consider

L79 Include also the total area of the modelled catchments, as well as the glacier area, for both regions. Include the year for this reference, if it's a website it can be when it was accessed.

L83 0.3% - check throughout that you use points not commas as the decimal separator

Figure 1 Can you include the modelled catchment areas as well? Maybe make the hydrometric station stars larger so they are more distinguishable from stakes. In the caption, 'various inset maps' rather than 'various zooms'.

Section 3.1 I feel like some if this is more modelling methods rather than data. It might be better to have a specific modelling section. I also feel like it is worth being more clear that S3M is a melt model only, it does not simulate discharge. I only say this since the discharge measurements are mentioned, which somehow hints that the model simulates discharge.

L102: 'second-order approximation' - this term has a very specific use, but I think you just mean that it's a reasonable simplifying assumption that glacier areas do not change and there is no debris cover.

L104: 'calibration in north-western Italy' - it would be good to give a little more detail on this, e.g. the parameters calibrated and whether they are varied spatially. Of course full details can be in related publications.

L110: I'm not completely sure what these %s represent.

L118-119 Did you consider using remote sensing datasets of glacier mass balance? E.g. Hugonnet et al. 2021 or other data specifically for Italy?

L132: 'every 500 m between 2000 m a.s.l. and 4500 m a.s.l.' would be more succinct

L138: Did you consider using drought indices? SPI etc.?

L146: 'we computed the correlation'

L149 'discharge that is'

L151 I think your simplified response to glacier contribution is ok, and reasonable given the data you have available, but it likely overestimates glacier contributions, since the discharge has accounted for losses to groundwater/ET for all inputs. Often in more complex hydrological models ice melt at least is not lost via ET or to groundwater (although it could be), but snowmelt more often is.

L156: Instead of 'concentration time' do you mean more the 'lag time' or 'travel time'

L163: You give the overall melt estimate biases, but the errors are still quite high at lower elevations, so I think this is worth mentioning. Ideally you could also include the mean bias overall for each region, and ideally also for each glacier (maybe figures could go in the SI).

Figure 2 It would be a good idea to include the glacier hypsometry and debris cover by elevation bins, then it's possible to see how important the glaciers are in each bin and where the debris cover is. Also add a horizontal line at 0 bias. Is the standard deviation of the comparisons? It might be useful just to see the actual modelled and measured mass balances, as then you can show the uncertainty of the measurements and model.

L171: Careful with commas in the temperature values here and below. Also, I would tend to only give air temperatures to 1 decimal point, since temperatures are rarely measured so accurately. Please also include the time period that the average was taken over for comparison.

L172-173: This text on the temperature differences is not so efficient, these values would be better as a graph of anomalies v elevation, you could have a line per month.

L175-177: Are all these temperature comparisons for June and July? Again, I think a figure would be better.

Figure 3 Consider if weekly averages might be clearer to interpret, mainly for the left hand panels.

L186 Write out 2023 if you want to start a sentence with it.

L193-209 – In general this paragraph could be more concise. It might work better to give the % differences in rainfall, rather than the comparison of the exact amounts, I think this might be clearer.

L205 'largely on average' – 'close to average' might be better

L210-213 The idea that winter and summer were the warmest seasons of 2022 is a bit confusing, as are the stats given here. Look at the mean per season anomaly (maybe per 3 month season) and then give the one season with the largest anomaly.

L214 It is mentioned that in 2023 the only clear difference compared to 2022 was that spring was noticeably cooler than the other seasons, but in 2022 the anomalies were much higher and the patterns between the seasons were different.

L231 What exactly do you mean here about melt out dates? The start of the date of any melt or the exposure of ice? The term 'melt out' is more used for instance when an ablation stake melts out as its no-longer in the ice, which is not what you mean I think.

L234-241 You could simplify the statistics given here for clarity, for instance give the range of the % change and the elevation pattern, e.g. the deficit ranged between -71% and -56%, decreasing with altitude. Since the actual values are in figures, you can afford for simplicity to simplify the values in the text.

L247 'More in details' is a bit odd phrasing, maybe 'In detail..' or 'Specifically..'

Figure 6 and paragraph L247-252, it might help understanding if you make clear again which place links to which river since in the text you give the outlet towns and in the Figure caption the rivers, just so the reader doesn't need to check back to the map again. It might also be useful in Figure 6 to add horizontal lines of the long term average, so the anomalies are really obvious. Also a comment in relation to this figure is that the discharges remain lower than normal, despite the increased glacier melt – does that mean that the glaciers are not able to fully compensate for the drought conditions?

L254 'This spur in' reads a bit strange, maybe 'This increase in'

Figure 7 it would be useful to have titles on the graphs with the places they represent. Also why are you showing only to week 39? There is still melt in both catchments. In the Figure 7 caption use 'upper' and 'lower' to indicate the plots, or (better) add letters to each panel. In fact this is a general comment, and adding letters to your figure panels throughout would be a good idea.

L267 'stood out clearly' is a little colloquial, plus I think the main point is that the glacier melt occurred earlier than usual. Maybe rephrase.

L271 Due to the importance of rainfall in interpreting the patterns it would be useful to include the precipitation patterns on the plots. I understand you might not want them to become too cluttered, but they could be small sub-plots underneath or light coloured bars. This would help the interpretation. You could also instead include the rainfall contribution to streamflow if you have this.

L286 'growing to 63 $m^3s^{-1}$'

L288 I do think your study does show these mechanisms of the impact of glacier melt on impacting droughts in mountain regions, but I don't know if the mechanisms themselves are so novel (as you point out with the previous research mentioned later in the discussion). However what is clear is the particularly significant role of glacier melt in these recent drought events, maybe this can be highlighted more here at the start of the discussion?

L299 ''previous studies (Huss and Hock, 2018…" just to add a space

L300 Maybe add some more studies on the point about glacier melt intensification during droughts

Figure 8 To help the last point in the discussion it might be useful to add the average long term Ta in the top panel, and average long term Pr in the middle panel. It would also be useful to show some breakdown of the rain/snow partitioning, if you have this information.

L303 'kept at a significant level' – could be clearer here – do you mean it remained significantly higher than the long term mean?

L313 Be careful here, the winter snow drought itself was not the cause of the warm and dry summer (excepting some link in the atmospheric conditions) but rather it resulted in reduced snow cover of the glaciers. I also understand your point that the melt would have been more radiation driven, but the link here is the albedo, which I would guess would be darker given earlier ice exposure. It might be worth to mention the albedo link here so its clear.

L315 Do you mean late winter snowfall? Just to be clear as it influences the discussion, maybe give the month and the amount of snowfall more than normal. Although I agree that summer snowfalls could be important for ice melt rates, the mechanisms as described here could be clearer, what is the effect of these snowfalls on the magnitude of melt rates and how does this compare to normal years?

L322 to 324 Can you expand on this point about compound droughts and heatwaves and how they relate to cascading cryosphere-hydrological droughts? Do you mean this is more of an autumn snow drought and/or more to do with autumn heat waves? I feel like there are some details missing here, it is because the lack of autumn snow means that glacier melt can become high as soon as there is a period of higher autumn temperatures? Or more because air temperatures are particularly high and the surface type is less important?

L330 I would tend to avoid the term 'second-order' in this case, rather likely glacier evolution within a relatively short period would likely only influence melt outputs by a small amount relative to the magnitude of melt variability due to climate conditions

L333-334 Maybe give some examples of possible models, e.g. a DETI melt model (Carenzo et al., 2016), and some examples of glacier evolution models which would be suitable (e.g. Huss et al., 2010, Jouvet and Cordonnier, 2023).

Carenzo et al. (2016) An enhanced temperature index model for debris-covered glaciers accounting for thickness effect, *Advances in Water Research*, 94, 457–469, http://dx.doi.org/10.1016/j.advwatres.2016.05.001

Huss et al. (2010) Future high-mountain hydrology: a new parameterization of glacier retreat, *Hydrol. Earth Syst. Sci*., 14, 815–829, doi:10.5194/hess-14-815-2010

Jouvet and Cordonnier (2023) Ice-flow model emulator based on physics-informed deep learning, *Journal of Glaciology*, 69(273), 13-26

L 340 and 241 – Are the glacier area changes really precise to 0.01%? Consider the precision of values given, they should reflect the precision of the measurements (please also check this throughout the manuscript).

L345 I am not sure if it is possible given your model set-up, but it would be a good test to do a run with the minimum glacier areas at the end of the period. You could therefore quantify the effect of the glacier area change on the overall results. Its ok to not evolve the glacier areas if this is not possible in the model, but knowing the extreme case would give you confidence that the effect is small enough as to not influence your results. I don't think the last point validates your results directly, maybe remove this last clause.

L359  'clear if one'

L360 'more than doubled from…'

Appendix: I think you could probably fit these figures in the main paper, for instance include A3 next to Figure 5, A2 could go in panels under the current Figure 4 and A1 could fit next to Figure 3.